# Relationship between Neuroticism, Spiritual Well-Being, and Subjective Well-Being in Korean University Students

**Jieun Yoo** [1], **Sukkyung You** [2,*] and **June Lee** [2]

1 Department of Christian Education, Anyang University, Anyang 1402822, Korea; msje9295@anyang.ac.kr
2 College of Education, Hankuk University of Foreign Studies, Seoul 270, Korea; junelee@hufs.ac.kr
* Correspondence: skyou@hufs.ac.kr

**Abstract:** Previous studies on mental health and quality of life have revealed that religiosity/ spirituality was positively associated with indicators of well-being and personality factors. However, limited research has examined the relationship between spiritual well-being, the subfactors of the personality factor *Neuroticism* (i.e., anxiety, hostility, depression, self-consciousness, impulsiveness, and vulnerability), and subjective well-being in a non-Western sample. The present findings revealed that the five subfactors of neuroticism did not have an equally negative or positive effect on spiritual and subjective well-being among Korean undergraduate University students. Regarding its subdimensions, vulnerability was strongly associated with spiritual well-being, while depression was closely linked to subjective well-being. Moreover, we found that spiritual well-being exerted significant effects on subjective well-being above personality factors. The significance of the findings and directions for further research have been discussed.

**Keywords:** spiritual well-being; subjective well-being; neuroticism; personality

## 1. Introduction

Issues related to living a happy life have long been the ultimate concern for humans, and numerous studies across several decades have investigated the topic of well-being. Recently, the Korean public has begun to pay more attention to well-being. The Korean government has placed its citizens' health and happiness as its top agenda and established policy priorities for their happiness (Cho 2018; Wang 2014). Despite the interest in well-being, according to the World Happiness Report 2020 on citizens' perceived level of happiness, Korea ranked 35th among the 37 countries of the Organisation for Economic Co-operation and Development (OECD) (Choi 2021). Among all age groups, those in their twenties reported the lowest level of happiness (Ministry of Health & Welfare 2020). People in their twenties experienced the highest level of negative emotions, such as boredom, anxiety, and depression, and felt the most unstable. They also showed the lowest level of self-esteem and the highest levels of materialism and comparison with others. This suggests the need for special attention on well-being and adaptation research.

Previous studies have revealed that personality factors are connected with various well-being indicators. Among the Five-Factor personality factors (i.e., Neuroticism, Extraversion, Openness, Agreeableness, and Conscientiousness), neuroticism has emerged as one of the most potent predictors of subjective well-being (Lucas and Diener 2015; Soto 2015). Additionally, neuroticism was also related to religiosity (Ehsan and Pournaghash-Tehrani 2012; Saroglou 2002) and spirituality (Sood et al. 2012). However, it is unclear whether neuroticism is associated with subjective and spiritual well-being among Korean young adults. Previous studies included primarily Caucasian samples, which has limited the generalizability to other populations. Therefore, it is necessary to investigate whether the personality factor of neuroticism would influence well-being among Korean undergraduate University students. Furthermore, we need to specifically examine how the subfactors

of neuroticism influence the diverse aspects of well-being. We need to study the detailed personality characteristics that predict well-being to contribute to the search for various measures to support individuals who suffer from low subjective well-being and enhance it.

*Background*

Subjective well-being is defined as a person's cognitive and affective evaluations of his or her life (Lucas and Diener 2015). Most individuals think that external conditions, such as work and income, are essential for happiness. However, abundant literature has provided evidence that personality factors are one of the steadiest influencers of subjective well-being (Lucas and Diener 2015). Concerning the Five-Factor personality factors, people who are more extroverted, agreeable, conscientious, open-minded, and emotionally stable are likely to experience more positive emotions, less negative emotions, and greater life satisfaction (Steel et al. 2008). Among the personality factors, neuroticism is of particular interest. General temperaments, such as neurosis, which generate more frequent experiences of negative emotions and personality characteristics, such as depression, are significant factors in predicting subjective well-being.

The Five-Factor personality factor *Neuroticism* is reported as the most critical negative predictor of subjective well-being (Lucas and Diener 2015). Neuroticism includes six subfactors which are anxiety, hostility, depression, self-consciousness, impulsiveness, and vulnerability (Costa and McCrae 1992). Among the sub-components of subjective well-being, neuroticism is particularly relevant to negative emotions (Miller et al. 2009). Other studies reported similar results, which suggested that individuals with high scores on neuroticism were likely to undergo negative emotions (i.e., anxiety, anger, sadness, and disgust) (Hiebler-Ragger et al. 2018). Thus, neuroticism influenced the emotional dimension of subjective well-being, comprising predominantly negative emotions (Lucas and Diener 2015).

Several studies have been conducted on the relationship between neuroticism and religiosity or spirituality (Henningsgaard and Arnau 2008; Johnstone et al. 2012). For instance, church-going members showed lower neuroticism levels than non-church-going members (Argyle 2001). Further, extrinsic religiosity was negatively related to neuroticism (Saroglou 2002). Concerning spirituality, Maltby and Day (2001) found that neuroticism was positively related to external/ritual spirituality among men. However, spirituality was negatively correlated to neuroticism (Saroglou 2002). Further, individuals who identified themselves as "spiritual but not religious" had a high level of neuroticism (Schnell 2012).

Mental health is defined as the absence of psychological distress indications and the presence of a flourishing psychological well-being. Furthermore, research on mental health and quality of life has revealed that spiritual well-being is linked with various indicators of an individual's physical and psychological health (Jo et al. 2019; Paloutzian et al. 2012; Yaghoobzadeh et al. 2018). Both religiosity and spirituality have been conceptualized as being multidimensional constructs (Van Cappellen et al. 2016). Spirituality, focusing on connectedness with sacred others (Saroglou 2014; Wuthnow 1998) or searching for the meaning of life in sacred ways (Pargament 1999), is generally separated from the institutionalized religion. Each religion approaches religiosity or spirituality differently. For example, it is usually considered that evangelical protestants concentrate utmost on the intimate connection with God, and Jews emphasize religious practices and community activities (Johnstone et al. 2012). The Buddhist way of understanding spirituality is different as it excludes a transcendent being and defines it as a purely internal quality or character (Harvey 2013). Therefore, given the complexity of religiosity and spirituality, this study defined the concept of spiritual well-being as one's awareness of the spiritual value of life wellness.

Extensive studies have confirmed the significant relationship of neuroticism with religiosity/spirituality and subjective well-being. Yet, limited studies to date have empirically examined the mechanism from neuroticism to spiritual and subjective well-being within the same model (Ramanaiah et al. 2001; Szczesniak et al. 2019; Unterrainer et al. 2010). Thus,

it is necessary to investigate the more complex mechanism between the personality factor *Neuroticism*, spiritual well-being, and subjective well-being, rather than simple bivariate relations. Moreover, limited empirical studies have explored this relationship using a non-Western sample. Therefore, more research is needed using a diverse sample. Additionally, extant research does not empirically examine how each subfactor of neuroticism influenced the different types of well-being. A more detailed evaluation of personalities explains subjective well-being more effectively depending on the personality characteristics. Thus, the current study intended to explore the effects of the six subdimensions on both spiritual well-being and subjective well-being. Furthermore, we also studied the mediating effects of spiritual well-being between neuroticism and subjective well-being.

Prior studies implied mediating mechanisms that connect personality factors and subjective well-being. Religiosity is one of the possible mediating factors given its motivational nature. Ellison (1991) claimed that the profits of religion were primarily cognitive since many religions provided an interpretive framework that assisted people in comprehending their own life experiences. Previous studies found that religious beliefs and positive religious coping were linked with a higher subjective well-being. For example, personal prayer and religious backing were substantial predictors of subjective well-being (You and Yoo 2016). Extant research has also provided evidence that religiosity and spirituality have a salutogenic role that allows individuals to find meaning in facing challenging life events (Cheon and Yoo 2019; Szczesniak et al. 2019; Unterrainer et al. 2010).

Two hypotheses have followed from the extant research.

**Hypothesis 1.** *Six subdimensions of neuroticism would predict lower levels of both spiritual well-being and subjective well-being.*

**Hypothesis 2.** *Spiritual well-being would mediate the relationships between neuroticism and subjective well-being.*

## 2. Methods

### 2.1. Participants

Five hundred university students were contacted to participate in this study in Seoul, Korea. Four hundred seventy-nine students agreed to participate in the survey (37.5% male). The mean age of the participants was 22.76 years (*SD* = 3.73 years). Sample characteristics are summarized in Table 1.

**Table 1.** Descriptive statistics.

|  | **Mean** | **SD** | **Range** |
|---|---|---|---|
| Individual background |  |  |  |
| Age | 22.76 | 3.73 | 17–55 |
| Male ∝ | 37.5 |  | 0–1 |
| Married ∝ | 5.2 |  | 0–1 |
| Education | 2.04 | 0.26 | 1–4 |
| Perceived poverty | 1.98 | 0.75 | 1–5 |
| Religious affiliation |  |  |  |
| Protestants ∝ | 61.7 |  |  |
| Catholic ∝ | 8.6 |  |  |
| Buddhists ∝ | 4.1 |  |  |
| No affiliation ∝ | 26.4 |  |  |
| Neuroticism subfactors |  |  |  |
| Anxiety | 3.11 | 0.69 | 1–5 |
| Hostility | 2.37 | 0.57 | 1–5 |
| Depression | 2.78 | 0.67 | 1–5 |
| Self-consciousness | 2.80 | 0.51 | 1–5 |
| Impulsiveness | 3.00 | 0.56 | 1–5 |
| Vulnerability | 2.81 | 0.55 | 1–5 |
| Well-being outcomes |  |  |  |
| Life satisfaction | 3.42 | 0.63 | 1–5 |
| Positive affect | 3.20 | 0.81 | 1–5 |
| Negative affect | 3.79 | 0.63 | 1–5 |
| Spiritual well-being | 3.75 | 0.80 | 1–6 |

Note: ∝ Dichotomous variables are described as proportions.

### 2.2. Procedures

After receiving institutional review board approval, data were collected during the Fall semester of 2019. All participants provided their informed consent for inclusion before they joined the study. During class, those who chose to participate were given the survey in a paper-and-pencil set-up. All procedures were conformed to institutional ethical guidelines for research on human subjects.

### 2.3. Measures

#### 2.3.1. Demographic Characteristics

To adjust for participants' demographic characteristics, we included gender, age, marital status, education, and perceived poverty as control variables.

#### 2.3.2. Well-Being Measures

**Subjective well-being.** Following previous studies (e.g., Lucas and Diener 2015; Schimmack 2008), we assessed subjective well-being using The Satisfaction with Life Scale (SWLS; Diener et al. 1985) and The Positive and Negative Affect Schedule (PANAS; Watson et al. 1988).

**Life satisfaction.** The Satisfaction with Life Scale (SWLS; Diener et al. 1985) was used to measure life satisfaction. The SWLS consists of five items (e.g., In most ways, my life is close to the ideal) on a five-point response scale (1 = *strongly disagree to 5 = strongly agree*). A higher score indicates greater satisfaction with life. The Cronbach's alpha coefficient for the current sample was 0.82.

**Positive and Negative Affect Schedule (PANAS).** The Positive and Negative Affect Schedule (PANAS; Watson et al. 1988) is a self-report assessment of youths' general emotional experience using a five-point response scale (1 = *strongly disagree to 5 = strongly agree*). This scale included a total of twenty items on positive affect (e.g., joyful, delighted, and cheerful) and negative affect (e.g., scared, lonely, and sad). The Cronbach's alpha coefficient for the current sample was 0.83.

**Spiritual Well-Being Scale (SWBS).** We measured spiritual well-being using The Spiritual Well-Being Scale (Paloutzian and Ellison 1991). SWBS is a twenty-item scale with two subscales, religious well-being (e.g., I believe that God or a higher power loves me and cares about me) and existential well-being (e.g., I feel that life is a positive experience). The Cronbach's alpha coefficient for the current sample was 0.77.

NEO Personality Inventory-Revised-Neuroticism Subscale (NEO-PI-R-N).

The NEO-PI-R (Costa and McCrae 1992) is a 48-item scale that assesses individual levels of neurotic attitudes (e.g., I often feel tense and jittery). The Cronbach's alpha coefficient for the current sample was 0.84.

### 2.4. Analysis

To examine the hypotheses regarding the relationships among neuroticism subscales and well-being measures, a hierarchical regression analysis was directed using Mplus (Muthén and Muthén 2006). The first model contained only individual contextual variables; the second model presented neuroticism subfactors. Both models assessed the effect size and statistical significance of a myriad of predictors concurrently; we decided on the unique contribution of each variable in the model controlling for the effects of the other variables involved.

The possible mediating effect of spiritual well-being in the association between personality factor subfactors and subjective well-being was verified using structural equation modeling (SEM) using Mplus. Model fit was evaluated based on several conditions: non-normed fit index (NNFI; Bentler and Bonett 1980), comparative fit index (CFI; Bentler 1990), and root mean square error of approximation (RMSEA; Steiger and Lind 1980). The NNFI and CFI are absolute fit indices, and values nearer to one indicate a perfect model. Conservatively, NNFI and CFI value larger than 0.90 shows a good model fit. The RMSEA

measures the degree of misfit and values lesser than 0.06 specify a good model fit (Hu and Bentler 1999).

## 3. Results

### 3.1. Descriptive Statistics

Table 1 shows the descriptive statistics, including means, standard deviations, and ranges of the variables in this study. Table 2 shows significant bivariate correlations among the neuroticism subscales and well-being measures.

**Table 2.** Correlations among variables.

| | 1 | 2 | 3 | 4 | 5 | 6 | 7 | 8 | 9 |
|---|---|---|---|---|---|---|---|---|---|
| 1. Anxiety | | | | | | | | | |
| 2. Hostility | 0.35 * | | | | | | | | |
| 3. Depression | 0.72 * | 0.43 * | | | | | | | |
| 4. Self-consciousness | 0.58 * | 0.33 * | 0.65 * | | | | | | |
| 5. Impulsiveness | 0.28 * | 0.48 * | 0.42 * | 0.38 * | | | | | |
| 6. Vulnerability | 0.64 * | 0.36 * | 0.70 * | 0.61 * | 0.45 * | | | | |
| 7. Life satisfaction | −0.50 * | −0.28 * | −0.70 * | −0.43 * | −0.30 * | −0.62 * | | | |
| 8. Positive affect | −0.33 * | −0.14 * | −0.53 * | −0.32 * | −0.18 * | −0.53 * | 0.79 * | | |
| 9. Negative affect | 0.56 * | 0.37 * | 0.65 * | 0.45 * | 0.39 * | 0.50 * | −0.67 * | −0.31 * | |
| 10. Spiritual well-being | −0.13 * | −0.24 * | −0.23 * | −0.15 | −0.14 | −0.28 * | 0.46 * | 0.43 * | −0.23 * |

Note: * $p < 0.05$.

### 3.2. Hierarchical Regression Analyses

The hierarchical regressions for well-being results have been presented in Table 3. Firstly, the demographic variables went into the model; however, these variables (i.e., age, gender, marital status, education, and perceived poverty) explained minimal variance in the results. After controlling for differences in demographics, all subfactors of neuroticism significantly impacted well-being outcomes, with a substantial increase in the proportion of explained variance.

**Table 3.** Hierarchical regression analysis predicting subjective and spiritual well-being outcomes.

| | Positive Affect | | Negative Affect | | Life Satisfaction | | Spiritual Well-Being | |
|---|---|---|---|---|---|---|---|---|
| | Model 1 | Model 2 | Model 1 | Model 2 | Model 1 | Model 2 | Model 1 | Model 2 |
| Individual background | | | | | | | | |
| Age | 0.04 | −0.01 | −0.13 | 0.11 | −0.04 | −0.05 | −0.05 | −0.05 |
| Gender (male = 1) | −0.01 | −0.07 | −0.08 | 0.04 | −0.07 | 0.05 | −0.16 * | −0.12 * |
| Marital status (married = 1) | 0.12 * | 0.05 | 0.09 | −0.06 | 0.05 | 0.02 | −0.04 | −0.001 |
| Education | 0.04 | 0.05 | 0.04 | −0.04 | −0.03 | −0.03 | −0.03 | −0.06 |
| Perceived poverty | −0.12 * | −0.11 * | 0.14 * | 0.13 * | −0.31 * | −0.27 * | −0.16 * | −0.11 * |
| Neuroticism subfactors | | | | | | | | |
| Anxiety | | −0.23 * | | 0.17 * | | 0.10 | | 0.18 |
| Hostility | | 0.08 | | 0.07 | | 0.02 | | −0.19 * |
| Depression | | −0.53 * | | 0.46 * | | −0.66 * | | −0.15 * |
| Self-consciousness | | 0.10 | | −0.03 | | 0.13 | | 0.05 |
| Impulsiveness | | 0.09 | | 0.10 | | 0.05 | | 0.09 |
| Vulnerability | | −0.42 * | | 0.01 | | −0.31 * | | −0.29 * |
| F (*df*) | 3.19(5) | 4.69(6) | 1.53(5) | 15.60(6) | 8.14(5) | 21.54(6) | 4.41(5) | 2.69(6) |
| $R^2$ ($\Delta R^2$) | 0.041(0.028) | 0.346(0.313) | 0.021(0.007) | 0.436(0.408) | 0.098(0.086) | 0.516(0.492) | 0.056(0.043) | 0.120(0.076) |

Note: * $p < 0.05$.

Specifically, for subjective well-being (see Table 3), individual demographic variables (Step 1) accounted for a significant but practically small amount of the variance in subjective well-being outcomes (*adjusted $R^2$* ranged from 0.02 to 0.09, $p < 0.05$). In Step 2, when neuroticism subfactors were included, the total model explained 35 to 52% of the variance in subjective well-being outcomes (*adjusted $R^2$* ranged from 0.35 to 0.52, $p < 0.05$). The results

showed that depression, followed by the vulnerability and anxiety facets of neuroticism, uniquely predicted subjective well-being. For spiritual well-being (see Table 3), individual contextual variables (Step 1) accounted for a significant but practically small amount of the variance in spiritual well-being (*adjusted $R^2$* = 0.06, *p* < 0.05). In Step 2, when neuroticism subfactors were entered, the total model explained around 12% of the variance in spiritual well-being outcomes (*adjusted $R^2$* = 0.12, *p* < 0.05). The results showed that vulnerability, followed by neuroticism's hostility and depression facets, uniquely predicted spiritual well-being.

### 3.3. Testing the Structural Equation Model

We verified the mediational model to evaluate the likelihood of the hypothesis that spiritual well-being mediates the relationship between neuroticism subfactors and subjective well-being. The results showed that the mediational model fit the data well. For the current sample, the mediational model produced an overall $\chi^2(405)$ value of 739.53, with CFI = 0.957, NNFI = 0.951, and RMSEA = 0.052. Figure 1 offers the standardized parameter estimates for this model.

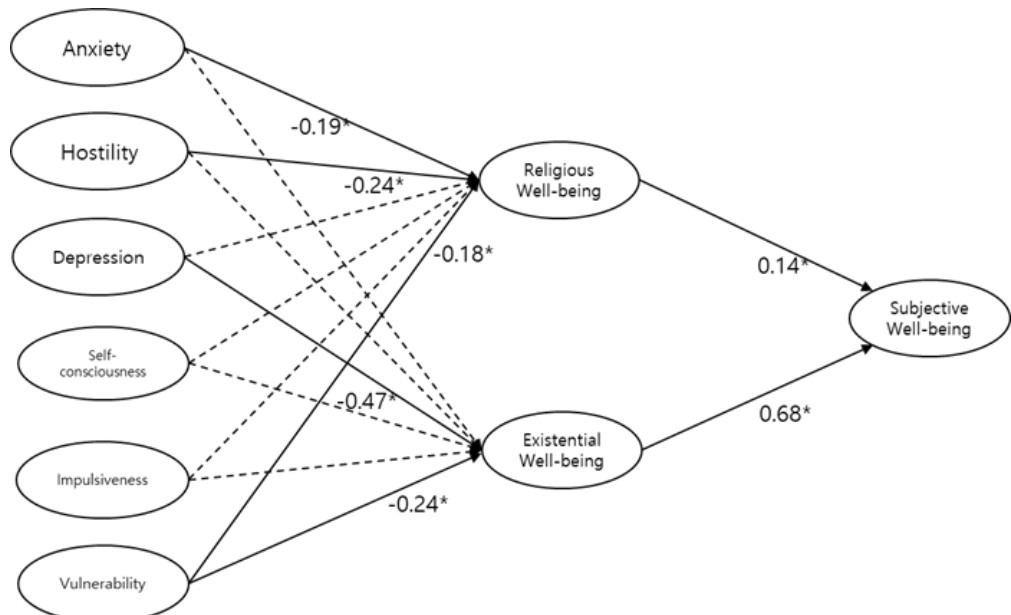

**Figure 1.** The structural equation model depicting the mediating effect of spiritual well-being on the association between subfactors of neuroticism and subjective well-being. Note. * *p* < 0.05.

### 4. Discussion

Our primary purpose was to inspect the connection between personality factor neuroticism, spiritual well-being, and subjective well-being among Korean undergraduate University students. First, the effects of the neuroticism on various well-being outcomes were examined using hierarchical regression analyses. Second, the mediating effect of spiritual well-being on the association between neuroticism and subjective well-being was examined using SEM. The results of the analyses are described below.

First, our findings showed that, among the subfactors of neuroticism, depression was the strongest predictor of subjective well-being (i.e., positive and negative emotions and life satisfaction). Specifically, individuals with higher levels of depression had lower life satisfaction and felt less positive and more negative emotions. However, this does not indicate that the other unpleasant elements of neuroticism were not linked with subjective well-being. Those who tended to experience more negative emotions, such as anxiety or vulnerability, were more likely to experience depression (Lee and Lee 2005; Schimmack et al. 2004). Additionally, depression was prevalent in Koreans, especially in those in their twenties. When compared with the figure in 2012, the number of patients with depression

increased by 22% in the year 2017 (Ministry of Health & Welfare 2018). Unlike the other age groups that showed a decrease in their level of depression, those in their twenties experienced a continuous increase (Ministry of Health & Welfare 2020). Depressive symptoms were also reported to be associated with suicide, which was ranked as a leading cause of mental-health-related deaths in Korea (Koo 2018). Therefore, this specific disposition of young Korean adults to experience increased depression is likely to be necessary for predicting subjective well-being. The efforts to alleviate depression and vulnerability, such as cognitive behavior programs, may positively influence increased well-being.

Second, this study revealed that vulnerability, hostility, and depression were predictors of spiritual well-being, and vulnerability exhibited the most substantial effect on spiritual well-being. In other words, Koreans with higher vulnerability exhibited a reduced spiritual well-being. Korea has gone through rapid economic development and changes in social values, which has left individuals vulnerable to stress, which causes mental ill-health (Koo 2018). As an outcome, mental health issues have become a national concern. Traditionally, coping strategies have included religious and spiritual activities, such as church attendance, which promote an internal locus of control and well-being in stressful situations (Lee et al. 2018; You and Yoo 2016). However, these coping strategies did not strengthen spiritual well-being, especially among vulnerable individuals when dealing with life stressors. Thus, religious leaders may encourage young adults to focus on spiritual interactions with a higher power, such as private prayer practices, to improve their spiritual well-being.

Third, our findings showed that spiritual well-being mediated the link between neuroticism and subjective well-being. Our results were consistent with those from previous studies. Specifically, we confirmed the positive association between spiritual well-being and subjective well-being (Pilger et al. 2017; Shahbaz and Shahbaz 2015; Unterrainer et al. 2010). This indicates that spiritual well-being is an integral factor of holistic wellness. The spiritual aspect is the central core that connects every other aspect, such as the physical, emotional, and social aspects. Subjective well-being is the cognitive and emotional reaction towards life experiences. To predict happiness, considering subjective factors matters more than evaluating objective factors. The current study confirms the extant research (Hwang et al. 2011; Unterrainer et al. 2010; You et al. 2019) concerning the positive association between spiritual well-being and subjective well-being and balances the gap between previous studies. Previous research employed regression analyses to test the relationship between the variables, thereby obscuring their causal relationship. However, this study utilized SEM and explored the directionalities of these relations.

This study has provided insights into the association between personality and mental health. Current findings suggest a new insight of empirical investigation where more comprehensive arrays of well-being, including spiritual well-being, can be examined within the Five-Factor personality factors. Particularly, neuroticism acts as a risk factor that has a detrimental effect on both spiritual and subjective well-being. Furthermore, in terms of novelty, the current study aided in the detailed assessment of neuroticism and a more profound explanation of spiritual and subjective well-being in Koreans. Previous studies have been conducted with higher-order personality dimensions of neuroticism. Thus, our findings provide more detailed information on personality and mental health by utilizing the subfactors of neuroticism that underlie the higher-order neuroticism factor. The concrete examination of the subdimensions of personal factors that have an essential effect on spiritual and subjective well-being is essential for predicting mental health. The current study's findings provide evidence for the consultation and guidance required by college students to increase their well-being and facilitate their adaptation to campus life. For instance, our findings suggest that experiencing positive emotions by taking courses and participating in student guidance activities, such as a consultation provided by the lecturers, will contribute to a higher level of college students' well-being. Moreover, counselors need to focus on their clients' problems by understanding their personality factors, such as the depression, anxiety, or vulnerability dimensions of neuroticism. This

would enable them to utilize efficient problem-solving strategies by considering the client's negative emotions and provide unique solutions to improve their life satisfaction.

## 5. Study Limitations

There are several limitations to this study. First, the present study was conducted on college students, and most were female (62.5%). Thus, this may restrict the generalization of the current findings. Second, cross-sectional data were collected from participants using one-time point self-reported questionnaires. Due to the study design, long-term relationships among variables cannot be formed. Third, there was a slight religious variance in the current study sample. Particularly, a majority of the participants came from a Christian religious background (62%) with a tiny Buddhist sample (4%). The spirituality of religion can be perceived differently among diverse major religious traditions (Cohen and Johnson 2017). Future studies should contain a greater diversity of religions to be more generalizable and avoid selection bias.

## 6. Conclusions

The current study contributed to the field of personality and mental health research. Specifically, it presented empirical evidence for the link between the subfactors of neuroticism and spiritual and subjective well-being among Korean young adults. Moreover, by relating spiritual well-being to personality factors, the current findings contribute to continuing discussion regarding the importance of approaching spiritual topics in the area of mental health and psychology of personality.

**Author Contributions:** Conceptualization, S.Y.; Data curation, J.Y.; Formal analysis, J.Y.; Funding acquisition, S.Y. and J.L.; Investigation, S.Y. and J.L.; Methodology, S.Y.; Resources, S.Y.; Validation, S.Y.; Writing—original draft, J.Y. and S.Y.; Writing—review & editing, S.Y. and J.Y. All authors have read and agreed to the published version of the manuscript.

**Funding:** This work was supported by the Hankuk University of Foreign Studies Research Fund 2022.

**Institutional Review Board Statement:** All procedures performed in studies involving human participants were in accordance with the ethical standards of the institutional and/or national research committee and with the 1964 Helsinki declaration and its later amendments or comparable ethical standards. The entire survey was reviewed and approved by a professional panel of educational specialists and school counselors (Project identification code No. CSED-2019-007A). Ethic Committee Name: Center for Social and Emotional Development, Hankuk University of Foreign Studies (Approval Code: CSED-2019-007A, Approval Date: 17 July 2019).

**Informed Consent Statement:** Informed consent was obtained from all subjects involved in the study.

**Data Availability Statement:** Not applicable.

**Conflicts of Interest:** The authors declare that they have no conflict of interest.

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
