# Peer review of "Relationship between Neuroticism, Spiritual Well-Being, and Subjective Well-Being in Korean University Students"

_religions, doi:10.3390/rel13060505_

Round 1
Reviewer 1 Report
This is a very interesting submission exploring the effects of the six sub-dimensions of neuroticism (anxiety, hostility, depression, self-consciousness, impulsiveness, and vulnerability) on both religious/spiritual well-being and subjective well-being as indicated by life satisfaction, positive affect, and lack of negative affect among Koreans (presumably South Koreans? This needs to be made more explicit in the submission). The study reported in this paper is important since there is limited research examining the relationship between religious/spiritual well-being, the sub-factors of personality factor Neuroticism and subjective well-being in a non-western sample. The paper is generally well-written, and the analysis is both thorough and detailed.
However, there are two concerns that I have with the submission, as it presently stands, that need to be addressed before it can be published. The first concern lies in the authors’ use of the term religious/spiritual. While there is an attempt on page 3 of the submission to distinguish between religion and spirituality, the paper tends to place both terms together, intimating that they are synonymous or at least closely connected. This is not the case and there is a plethora of academic literature that outlines in some detail the distinction between these two terms. It is possible for an individual to regard her or himself as being ‘spiritual’ but not religious. The authors in fact note this, but do not go on to discuss the possible implications here. A greater attempt needs to be made in this paper to distinguish between ‘religion’ (‘religiosity’) and ‘spiritual’ (‘spirituality’). Such a distinction may have implications for the findings. In any case, the distinction needs to be made in order to acknowledge the large body of literature that discusses such a distinction. I would not expect to see a lengthy exposition inserted into a revision, but I would argue that the distinction does need to be made in greater detail. (The authors also use the term ‘spiritual intelligence’ on page 2 of the submission. This is a contested term in the literature, and this should also be at least noted).
Similarly, the concept of ‘well-being’ is not defined, aside from a brief line in relation to its connection with happiness and satisfaction on page 2 of the submission. ‘Well-being’ is not a universal concept – it can mean different things in different social and cultural contexts. This needs to be recognized and made clear in the submission, and a short discussion as to how it is being used in this paper, with reference to the academic literature, needs to be included.
The second concern that I have is that it is not until the Methods section on page 4 of the submission that the reader is alerted to the fact that the sample is comprised solely of university students whose mean age is 22, or thereabouts. This is problematic because it is not reflected in the aim of the study, nor in many of the subsequent claims that are made in the light of the findings. For example, the statement on page 8 of the submission under the heading “Discussion” says “Our primary purpose was to investigate the relationship between personality factor 72 neuroticism, religious/spiritual well-being, and subjective well-being.” It should say, “Our primary purpose was to investigate the relationship between personality factor 72 neuroticism, religious/spiritual well-being, and subjective well-being among undergraduate University students” (or something similar). It needs to also be clear in the abstract and the introduction to the paper that the study explores the effects of the six sub-dimensions of neuroticism on both religious/spiritual well-being and subjective well-being as indicated by life satisfaction, positive affect, and lack of negative affect among Korean undergraduate University students. This may also have implications for some of the claims made in the findings section, where the claims appear to be made in relation to a wider range of age groups. The authors will need to look closely to see whether any amendments need to be made throughout the paper in the light of this.
Author Response
Reviewer1
This is a very interesting submission exploring the effects of the six sub-dimensions of neuroticism (anxiety, hostility, depression, self-consciousness, impulsiveness, and vulnerability) on both religious/spiritual well-being and subjective well-being as indicated by life satisfaction, positive affect, and lack of negative affect among Koreans (presumably South Koreans? This needs to be made more explicit in the submission). The study reported in this paper is important since there is limited research examining the relationship between religious/spiritual well-being, the sub-factors of personality factor Neuroticism and subjective well-being in a non-western sample. The paper is generally well-written, and the analysis is both thorough and detailed.
As suggested, we have added the sample information mentioned above.
However, there are two concerns that I have with the submission, as it presently stands, that need to be addressed before it can be published. The first concern lies in the authors’ use of the term religious/spiritual. While there is an attempt on page 3 of the submission to distinguish between religion and spirituality, the paper tends to place both terms together, intimating that they are synonymous or at least closely connected. This is not the case and there is a plethora of academic literature that outlines in some detail the distinction between these two terms. It is possible for an individual to regard her or himself as being ‘spiritual’ but not religious. The authors in fact note this, but do not go on to discuss the possible implications here. A greater attempt needs to be made in this paper to distinguish between ‘religion’ (‘religiosity’) and ‘spiritual’ (‘spirituality’). Such a distinction may have implications for the findings. In any case, the distinction needs to be made in order to acknowledge the large body of literature that discusses such a distinction. I would not expect to see a lengthy exposition inserted into a revision, but I would argue that the distinction does need to be made in greater detail. (The authors also use the term ‘spiritual intelligence’ on page 2 of the submission. This is a contested term in the literature, and this should also be at least noted).
As suggested, we have clarified the term religious/spiritual.
Similarly, the concept of ‘well-being’ is not defined, aside from a brief line in relation to its connection with happiness and satisfaction on page 2 of the submission. ‘Well-being’ is not a universal concept – it can mean different things in different social and cultural contexts. This needs to be recognized and made clear in the submission, and a short discussion as to how it is being used in this paper, with reference to the academic literature, needs to be included.
As suggested, we have added the definition of well-being.
The second concern that I have is that it is not until the Methods section on page 4 of the submission that the reader is alerted to the fact that the sample is comprised solely of university students whose mean age is 22, or thereabouts. This is problematic because it is not reflected in the aim of the study, nor in many of the subsequent claims that are made in the light of the findings. For example, the statement on page 8 of the submission under the heading “Discussion” says “Our primary purpose was to investigate the relationship between personality factor 72 neuroticism, religious/spiritual well-being, and subjective well-being.” It should say, “Our primary purpose was to investigate the relationship between personality factor 72 neuroticism, religious/spiritual well-being, and subjective well-being among undergraduate University students” (or something similar). It needs to also be clear in the abstract and the introduction to the paper that the study explores the effects of the six sub-dimensions of neuroticism on both religious/spiritual well-being and subjective well-being as indicated by life satisfaction, positive affect, and lack of negative affect among Korean undergraduate University students. This may also have implications for some of the claims made in the findings section, where the claims appear to be made in relation to a wider range of age groups. The authors will need to look closely to see whether any amendments need to be made throughout the paper in the light of this.
As suggested, we have added the sample information earlier in the Introduction and added sample information as mentioned.
Reviewer 2 Report
- There is no short characteristics of the research group in the abstract.
-
In the section entitled 1.1 Background the sentence The Five-Factor personality factor Neuroticism is reported as the most critical predictor of subjective well-being requires the following clarification: neuroticism is a negative predictor of well-being.
-
The paragraph concerning the importance of neuroticism for well-being begins with the Big Five personality model. In the same paragraph the author/authors refer to the research of Eysenck & Eysenck, who represent a different approach to personality than the Big Five. This paragraph needs clarification and ordering.
-
Participants and Measures are essentially well and comprehensively characterised. In the description of NEO-PI-R-N a sample statement is worth giving, just like in the characteristics of other research tools. In my viewpoint, the sentence at the end of the scale description: There are 12 significant associations between this measure and related variables, such as anger and 13 anxiety in college students (Choi et al. 2006) is not necessary and it can be deleted.
-
The results are clearly and well-presented.
-
The discussion is orderly, essentially correct and interesting.
-
The limitations of the research are the last part of the discussion. This fragment is worth distinguishing as a separate part of the paper.
- Conclusions are short but sufficient.
Author Response
There is no short characteristics of the research group in the abstract.
As suggested, we have added the sample characteristics in the Abstract.
In the section entitled 1.1 Background the sentence The Five-Factor personality factor Neuroticism is reported as the most critical predictor of subjective well-being requires the following clarification: neuroticism is a negative predictor of well-being.
As suggested, we have revised the sentence mentioned above.
The paragraph concerning the importance of neuroticism for well-being begins with the Big Five personality model. In the same paragraph the author/authors refer to the research of Eysenck & Eysenck, who represent a different approach to personality than the Big Five. This paragraph needs clarification and ordering.
As suggested, we have revised the sentence mentioned above.
Participants and Measures are essentially well and comprehensively characterised. In the description of NEO-PI-R-N a sample statement is worth giving, just like in the characteristics of other research tools. In my viewpoint, the sentence at the end of the scale description: There are 12 significant associations between this measure and related variables, such as anger and 13 anxiety in college students (Choi et al. 2006) is not necessary and it can be deleted.
As suggested, we have deleted the sentence mentioned above.
The results are clearly and well-presented.
The discussion is orderly, essentially correct and interesting.
The limitations of the research are the last part of the discussion. This fragment is worth distinguishing as a separate part of the paper.
As suggested, we have separated the limitation section.
Conclusions are short but sufficient.